# Behavior of KCNQ Channels in Neural Plasticity and Motor Disorders

**DOI:** 10.3390/membranes12050499

**Published:** 2022-05-06

**Authors:** Som P. Singh, Matthew William, Mira Malavia, Xiang-Ping Chu

**Affiliations:** Department of Biomedical Sciences, School of Medicine, University of Missouri-Kansas City, Kansas City, MO 64108, USA; somsingh@umkc.edu (S.P.S.); mrwmxd@umkc.edu (M.W.); mcmkp8@umkc.edu (M.M.)

**Keywords:** KCNQ channels, neural plasticity, pain, motor disorders, neurodegenerative disease

## Abstract

The broad distribution of voltage-gated potassium channels (VGKCs) in the human body makes them a critical component for the study of physiological and pathological function. Within the KCNQ family of VGKCs, these aqueous conduits serve an array of critical roles in homeostasis, especially in neural tissue. Moreover, the greater emphasis on genomic identification in the past century has led to a growth in literature on the role of the ion channels in pathological disease as well. Despite this, there is a need to consolidate the updated findings regarding both the pharmacotherapeutic and pathological roles of KCNQ channels, especially regarding neural plasticity and motor disorders which have the largest body of literature on this channel. Specifically, KCNQ channels serve a remarkable role in modulating the synaptic efficiency required to create appropriate plasticity in the brain. This role can serve as a foundation for clinical approaches to chronic pain. Additionally, KCNQ channels in motor disorders have been utilized as a direction for contemporary pharmacotherapeutic developments due to the muscarinic properties of this channel. The aim of this study is to provide a contemporary review of the behavior of these channels in neural plasticity and motor disorders. Upon review, the behavior of these channels is largely dependent on the physiological role that KCNQ modulatory factors (i.e., pharmacotherapeutic options) serve in pathological diseases.

## 1. Introduction

Ion channels serve as an aqueous conduit for several nuanced cellular processes to maintain the homeostatic direction of the body. Moreover, there are over 400 genes that encode for at least one ion channel subunit [1,2]. The various mechanisms for alternative splicing make for an enormous variety of subunit combinations designed for appropriate physiological functions. Among these, the largest and most diverse group of ion channels are potassium (K^+^) channels [2,3]. These channels are composed of tetrameric integral membrane regions, which form an aqueous pore for K^+^ to permeate across the membrane. This ion serves a critical role in maintaining electrical gradients during the repolarization of action potentials and maintaining the negative resting membrane potential [3,4].

Voltage-gated potassium channels (VGKCs, also Kv) form a broad distribution of channels in the nervous system as well as other tissues. Structurally, Kv channels are also a tetramer integral membrane pore-forming alpha subunit but also contain six transmembrane segmental helices, classified as S1–S6. In addition, the S1–S4 transmembrane segmental helices compose the actual voltage sensation region, and the latter two (S5–S6) units are the actual gate of the channel, as depicted in Figure 1. The voltage sensation region (S1–S4) is supple in its ability to adapt to shifting membrane potentials by creating a conformational shift. This shift spreads through the pore-forming subunit via interactions with the S4 transmembrane segments. In addition, this segment is also protected during depolarization of the action potential (AP). This protection is due to the presence of the acidic residues on S1 and S2 transmembrane segments, which limits deterrence [3,4,5].

Within the family of Kv channels, there are subfamilies that can be grouped according to the N- and C-terminal domains and encoded genes [5,6]. The importance behind the subfamily grouping lies in the Kv proteins, which can be functionally divergent with different membrane sensitivity potentials, gating interactions, and dynamic responses [4]. These subfamilies of Kv channels are all encoded by 40 genes, and current literature establishes exactly 12 subfamilies of Kv channels as a product of this gene encoding (e.g., Kv1–12) [6].

Historically, some of the earliest studies on voltage-gated ion channels (VGICs) were on the contemporary Kv7 subfamily [5,6]. Moreover, the understanding of the Kv7 subfamily was not immediate upon discovery. Rather, the literature initially focused on a concept known as the M channel. This channel was initially termed due to its activity as a low-threshold non-inactivating K^+^ channel [7]. They were named “M channels” as such because of pilot literature that showcased their inhibition via muscarinic acetylcholine receptors (mAChR) stimulation [5]. Today, the subunits of the subfamily Q Kv7 K^+^ (KCNQ) channel family are now known to be part of M channels and are a key target as the basis for pharmacological treatment modalities for a broad spectrum of neurological disorders. This is because Kv7 have been shown to be stimulated by membrane potentials that are more negative than the AP threshold due to their activity as a low-threshold non-inactivating K^+^ channel [5,6,7].

Structurally, the KCNQ channels are similar to their Kv channel relatives (Figure 1). However, the emphasis on these channels is in their ability to utilize their glycine residues to contribute to a major part of their K^+^ ion preference [8,9]. Specifically, the channels have glycine residues which utilize their carbonyl oxygen branches to form a shell that is specific for the size of K^+^ ions compared to Ca^2+^ and Na^+^ ions [9,10].

The KCNQ channels are responsible for the M currents during physiological processes, which is important in the regulation of various neuronal excitability [10]. The basis of which is formed by several different KCNQ isoforms forming heterotrimeric channels. The M-current is a non-inactivating sub-threshold current [9,10]. The increases in neuronal excitability have resulted from physiological modulation, pharmacological inhibition, and genetic mutations that affect the M-current [9,10,11]. The Kv7 channels can transiently induce the suppression of the M-current such that they limit the firing frequency of neurons [10,11]. Furthermore, it is the Kv7.2 and Kv7.3 channels which are specifically involved in the regulation of M-current, and some other channels can also play minor contributory roles [10,11,12,13].

With regards to the actual opening and closing of the KCNQ channel, there are several mechanisms. For example, KCNQ channels can open via binding of the phosphatidylinositol 4,5-bisphosphate (PIP2) ligand. The direct binding of gamma-aminobutyric acid (GABA) to the KCNQ channel can directly increase the likelihood that a KCNQ channel will open and allow K^+^ permeation. This mechanism seems to be GABA-specific as such a conformation has not been identified in KCNQ channels activated by other means. Secondly, inositol 1,4,5-trisphosphate (IP3)-mediated intracellular calcium signals promote PIP2 synthesis and, via calmodulin, will suppress the M-current [14,15]. In regard to neuronal KCNQ channels, their importance lies in the ability to modulate neurotransmitter release and somatic excitation in the nervous system. Robust production of PIP2 via hydrolysis agonizes four receptors in the sympathetic neurons of the superior cervical ganglion (e.g., M1, AT1, B2, and P2Y). Modulation of this system occurs via competitive or allosteric regulation of the membrane transport protein affinities for PIP2 molecules [1,5,6,7,8,9,10,11,16,17,18,19,20,21].

With this array of physiological properties found in KCNQ channels, there has been a growth in the literature on KCNQ channel property modifications for therapeutic treatment modalities, as well as the role of these channels in various pathological processes. Specifically, the alteration or loss of function (LOF) by these KCNQ (i.e., channelopathies) highlight their importance in physiological function in the body.

There are various phenotypic presentations of these channelopathies as most are due to genetic etiology amongst whichever genes are involved and the location of the channels, as depicted in Table 1 [17,18,19,20,21]. The most common genes involved in channelopathies are KCNQ1-5 (without consideration of spliced variants) [14,17,21]. KCNQ1 is most expressed in cardiac and cochlear tissue [14,22]. Specifically, cardiac KCNQ1 LOF mutations are associated with type 1 long-QT syndrome [22,23,24,25,26,27,28,29,30]. Cochlear KCNQ1 pathology involves the autosomal recessive long-QT syndrome (Jervell Lange-Nielsen syndrome), which is associated with potassium channelopathy leading to bilateral sensorineural hearing loss as well as the cardiac arrhythmia [26,27,28,29]. KCNQ2 is most expressed in the fetal cerebellum, hippocampus, and medulla [30,31]. Genetic mutation in KCNQ2 is often associated with benign familial neonatal seizures and early-onset epileptic encephalopathy [9,32,33,34,35,36,37,38,39,40,41,42,43,44,45]. KCNQ3 is also most expressed in the fetal cerebellum, hippocampus, and medulla [9,30]. In addition, KCNQ3 mutations are often associated with channelopathies in conjunction with KCNQ2 [33], but additional literature also supports KCNQ mutations in bipolar disorder [36] and various thyroid disorders [37]. Similar to KCNQ1 expression, KCNQ4 is most expressed in the cochlear hair cells but also in trigeminal ganglia [14]. This plays a key role in maintaining the K^+^ gradient for channel mechanosensation to carry K^+^ into hair cells to stimulate auditory sensation [14,38]. KCNQ4 mutations are often associated with auditory hearing loss and have therefore been a key target in developing pharmacotherapeutic options for hearing loss [39,40,41,42,43,44,45,46,47,48,49,50,51,52,53,54]. KCNQ5 is most expressed in neural tissue, including the retinal pigment epithelium [48]. However, the lack of recent literature on the profile of these encoded channel subfamilies suggests that there may be unknown channelopathies related to vision homeostasis [14,48]. The expression of these genes is more often in association with other KCNQ genes than what was separately outlined. In addition to KCNQ5, KCNQ1 and KCNQ4 are also often encoded to channels in the neuronal retina and may also have a degree of contribution to its physiological function [9,14,30]. Despite this, the importance of highlighting single gene encoding remains key to approaching neural pathophysiology [2,8,14]. Given this importance, the aim of this review is to provide an up-to-date understanding of the contemporary work of KCNQ channels in order to provide greater emphasis on KCNQ’s involvement in various pathophysiological processes distributed throughout the human body.

## 2. Modulation of Synaptic Plasticity by KCNQ Channels

There has been a greater development in the role of KCNQ channels among neuronal networks in the past decade. This has led to its consideration for potential pharmacotherapeutic applications [14]. The ability for neuronal modification, or neural plasticity, is a key area of focus in understanding the foundations of learning and memory functions. Anatomically, the origin of the literature on neural plasticity can be further refined by discussing the concept of synaptic plasticity. This concept focuses on hippocampal formation and two principal cell types: pyramidal neurons and granular cells. Specifically, the pyramidal neurons are composed of diverse branching of dendritic neurons, which are responsible for synaptic communication with other neurons [49,50]. The morphological formation of these neurons within the hippocampus leads to the further subfield classification of pyramidal cells in what is known as Cornu Ammonus (CA), divided into CA1, CA2, and CA3 [49,50]. These regions serve an important role in localizing KCNQ channel function in synaptic plasticity [49,51,52,53,54,55,56,57,58,59,60,61]. Within synaptic plasticity, two major models involved in the application of neural plasticity are long-term potentiation (LTP) and long-term depression (LTD) [59]. These models are activity-dependent, and the literature establishes their role in namesake enhancement or reduction in synaptic efficiency. Historically, LTP was initially found in animal models, which found a sustained enhancement in the hippocampus following high-frequency electrode stimulation. LTD was later recognized after laboratory models found the opposite effect following low-frequency simulations [61,62,63,64,65]. At the cellular level, the literature suggests there are numerous factors that play a role in creating the genres of synaptic efficiency and, ultimately, neural plasticity [63,64,65,66,67,68].

Historically, the literature establishes high concentrations of KCNQ2–5 channels in the perisomatic CA1 hippocampal regions [50]. Within dendritic CA1 regions, the current generated by KCNQ channels may not serve as robust of a role as seen in pyramidal CA1 regions [14,50]. Moreover, it has been seen that modulation of KCNQ currents via linopirdine and XE991 do not create effects on synaptic excitability. Rather, it has been shown that the axonal KCNQ channels create a backpropagation into the dendritic CA1 regions [14,65,66,67,68]. This may suggest that the quantity of KCNQ channels in the dendrites does not play as robust of a role in synaptic excitability as the axonal KCNQ channels do themselves [64,65,66,67,68]. This makes axonal KCNQ channels the greater focus of study.

It was initially found that pharmacologic KCNQ channel inhibition via linopirdine reduced spike frequency adaptation (SFA) in CA1 pyramidal neurons in vitro, but only after the initial spike [49,50]. Following this initial discovery, it was also found that KCNQ channel modulation also plays a role in after hyperpolarization, which ultimately supports the notion that KCNQ channels contribute to AP [66]. In addition, muscarinic channel inhibition (i.e., KCNQ) has been shown to stimulate an array of homeostatic neuroplastic changes in synaptic efficiency. This array, in combination with KCNQ’s contribution to AP, may suggest that this array occurs at different time points, which allows for understanding that a temporal process of these neuroplastic changes occurs rather than a synced process in the hippocampus [67]. If the behavior of KCNQ channels occurs in a temporal process, this can make way for a greater understanding of the role of KCNQ via LTP and, therefore, memory development.

LTP genre can be categorized as either dependent or independent of N-methyl-D-aspartate (NMDA) receptors. Within the NMDA receptor-dependent form of LTP, it is suggested that KCNQ inhibition via XE991 stimulates the opening of NMDA receptors mediated channels during LTP by stimulating the depolarization after AP firing when performed via theta-burst stimulation [65]. This behavior may suggest that XE991 inhibition could serve a pharmacotherapeutic role in improving memory. However, the literature regarding the modulation of KCNQ channels regarding memory and LTP is still not completely understood [62,63,64,65,66,67,68]. With regard to LTP in the presence of acute stress, it is well understood that stress impairs spatial memory retrieval. Flupirtine-induced activation of KCNQ channels in the CA1 region is found to have a neuroprotective effect on spatial memory retrieval in the case of acute stress. The mechanism behind these protective effects is suggested to be through the Akt/GSK-3β and Erk1/2 signaling pathways, and animal models have shown flupirtine treatments resulted in decreased expression of apoptosis factors (i.e., Bax) and upregulation of hippocampal p-Erk1/2 [66,67,68]. Likewise, literature establishes beneficial effects on memory via KCNQ pharmacological inhibition as well. In addition to the aforementioned effects on LTP by XE991, this inhibitor has also improved cognitive impairment secondary to acetylcholine (ACh) depletion animal model induced by the neuroexcitatory kainic acid [64,65,66,67,68]. The discovery of this behavior has led to a growing body of literature on potential therapeutic applications in Alzheimer’s Disease due to its nature as a cholinergic deficiency-related cognitive impairment [66,67]. In addition, the inhibition of KCNQ channels via linopirdine is also well-established in enhancing cognition via increased ACh release [63,65,66,67,68,69,70,71,72,73,74,75,76,77].

In contrast, while LTP typically occurs after a brief high-intensity stimulation of a postsynaptic neuron, LTD can be caused by prolonged low-intensity stimulation or simulation that occurs after the firing of an AP [69]. This leads to insufficient depolarization due to this lower level of stimulation. This does not generate a removal of the magnesium blockage of the NMDA receptor [78,79,80,81,82]. However, there is evidence that this stimulation is enough to open some NMDA receptors to allow for calcium ions into the cell. These cellular calcium levels are thought to activate a cellular cascade to remove α-amino-3-hydroxy-5-methyl-4-isoxazolepropionic acid (AMPA) receptors [77,78,79,80,81]. This reduces the postsynaptic glutamate receptor density, which decreases synapse efficiency and, therefore, memory and learning development. Despite the developments in literature dedicated to LTD, there is little literature on the effects that KCNQ channels have on this mechanism compared to LTP.

Other than the pharmacological inhibition of KCNQ channels, the inhibition by genetic proxy also serves a role in neural plasticity with regard to cognition. Animal models have shown epileptic seizures in addition to cognitive spatial memory impairment in cases of mutant or LOF genes that encode for KCNQ2 [82,83,84,85,86,87,88,89]. These effects brought upon by genetic inhibition challenge the protective and cognitive improvements seen in pharmacologic forms of KCNQ inhibition. This is where the literature ought to focus in order to determine if secondary factors that are not understood in these animal models also contribute to the memory impairment (i.e., hippocampal morphology) [90,91,92,93,94,95]. The epileptic phenotype, in conjunction with the cognitive impairment of these genetic models, may suggest additional psychomotor exploration [95,96,97,98,99,100,101,102,103,104].

Clinically, neuroplastic changes to the central nervous system are well documented in the case of chronic pain [105,106,107,108]. Moreover, there is a large number of literatures that focuses on the ability to enhance positive neuroplasticity as a clinical application for chronic pain treatment. Potassium channels can be regulated for the function of membrane excitability. The dorsal horn neurons and the DRG sensory neurons express the neuronal Kv7 channels, which are formed by Kv7.2, Kv7.3, and Kv7.5 subunits [80]. Depolarization of the resting membrane potential and increasing firing of nociceptive neurons occurs when Kv7 channels have either been blocked or have experienced a LOF. As such, a potential treatment for chronic pain can be increasing the function of such Kv7 channels in nociceptors. Retigabine, a Kv7 opener, was initially investigated as a potential seizure treatment but has been discontinued for concerns of toxicity and skin/retinal discoloration [109,110,111]. Recently, SCR2682 has been shown to decrease DRG neuron excitability in vitro and strongly activate Kv7 currents in neurons. In persistent, inflammatory, neuropathic pain models, SCR2862 activation of Kv7 currents reduced the thermal hyperalgesia and mechanical allodynia [112].

In addition to Retigabine and XE991, the anti-inflammatory painkillers celecoxib and diclofenac could potentially impact Kv7 channels, underlining the importance of Kv7 as a potential analgesic target [84]. Diclofenac can directly depress spinal nociceptive transmission and spinal reflexes. The literature by Baz et al. has emphasized that both diclofenac and flupirtine spinal effects are mediated through KCNQ channels [111,112,113,114,115]. Another study investigated the anti-nociceptive effects on chronic and inflammatory pain of the novel opener QO58-lysine on KCNQ channels, finding that there is a pharmacological effect of QO58-lysine effect on pain [85,86]. Inflammatory pain was reversed by administration of QO58-lysine without toxicity as well [85,86,87,88,89].

## 3. KCNQ Channels in Dopaminergic Motor Disorders

KCNQ channels have become important targets in disorders regarding dopaminergic pathways. Channels of this group in the ventral tegmental area (VTA) of the mesolimbic dopaminergic tract have been explored as targets of pain attenuation [80]. While this avenue of study has proven to be a promising utilization of these K^+^ channels in dopaminergic pathways, they are perhaps better associated with the treatment of motor disorders involving the neurotransmitter dopamine. Parkinson’s disease (PD) is a neurodegenerative disorder commonly associated with the loss of dopaminergic neurons in the substantia nigra pars compacta (SNc) [90,91,92,93,94,95,96,97,98,99,100,101,102,103]. As a result, the loss of nigrostriatal dopamine is associated with the motor symptoms classically observed in PD, such as muscular rigidity, bradykinesia, and tremor [84,85,86,87,88,89,90,91,92,93,94,95]. Recently, it has been suggested that the modulation of certain K^+^ channels is able to elicit protective effects upon nigrostriatal dopaminergic neurons, attenuating the motor symptoms observed in PD. Among the K^+^ channels capable of attenuating the motor-related symptoms of PD is the KCNQ family of K^+^ channels. KCNQ2 and KCNQ4 specifically have been shown to exist in high concentrations in the SNc compared to other members of the KCNQ family [95].

Activators of the KCNQ channels in the substantia nigra, such as retigabine (a KCNQ2-5 opener), can induce an inhibitory control of nigrostriatal neurons through increased K^+^ conductance [96,97]. This effect is able to attenuate the activity of dopaminergic neurons, diminishing the amount of dopamine released in the SNc. Conversely, the blockade of the KCNQ channels via XE991 had the opposite effect. Through the inhibition of K^+^ conductance in the dopaminergic neurons of the SNc, XE991 can induce greater excitability in these neurons [80]. It has therefore been hypothesized that XE991 may be a potential pharmacotherapeutic option in the treatment of PD. By targeting KCNQ channels in the SNc, increasing the generation of action potentials in existing dopaminergic neurons may potentially alleviate the symptoms caused by a loss of dopaminergic neurons, as seen in PD. A recent study conducted by Chen et al. attempted to determine the efficacy of XE991 in attenuating the motor symptoms associated with PD in vivo. Catalepsy was induced in rats treated with intraperitoneal injections of haloperidol to create a PD-like state. The study found that administration of XE991 into the SNc of the rats was able to alleviate the symptoms of catalepsy induced by systemic administration of haloperidol, providing an in-vivo example of the treatment of motor symptoms in a PD-like state via targeting of KCNQ channels [98].

In addition to increasing the excitability of dopaminergic neurons in the SNc, XE991 has been shown to provide neuroprotective effects as well by preventing dopaminergic neuronal death. The potent neurotoxin, 6-hydroxydopamine (6-OHDA), has been shown to replicate an irreversible state similar to PD by the destruction of dopaminergic neurons in the SNc [98,99]. A recent study conducted by Liu et al. attempted to observe the neuroprotective effects of XE991 on 6-OHDA treated rats [80]. During the study, it was found that rats with 6-OHDA injections in the medial forebrain bundle (MFB) exhibited significantly lower levels of tyrosine hydroxylase positive neurons in the SNc, compared to rats treated with a sham injection in the MFB. Additionally, another group of rats was treated with 6-OHDA as well as XE991 to observe the potential neuroprotective effects of blocking KCNQ channels. It was found via immunofluorescence that the survival ratio of tyrosine hydroxylase positive neurons was significantly greater in the group treated with XE991 and 6-OHDA (54.43%) than in the group treated only with 6-OHDA (16.63%), leading to the conclusion that XE991 was able to prevent attenuation of nigrostriatal dopamine in a PD-like state [80].

The neuroprotective effects associated with the blockade of KCNQ channels by XE991 may potentially be associated with an attenuation of motor symptoms associated with PD. In an open-field test, 6-OHDA injected mice treated with XE991 exhibited significantly greater walking speeds than mice treated only with 6-OHDA, displaying the ability of XE991 to counteract bradykinetic symptoms. A balance beam test designed to assess balance and coordination was also conducted, with higher scores representing a greater ability to traverse a balance beam. It was once again found that 6-OHDA injected rats treated with XE991 displayed significantly greater scores than their counterparts only injected with 6-OHDA. Finally, the study conducted a grip test in which rats who were able to hold onto a wire for a longer period of time exhibited greater muscular rigidity, a classic symptom of PD. The study found a significant decrease in grip time (and therefore, muscular rigidity) in rats injected with 6-OHDA treated with XE991 compared to rats only injected with 6-OHDA. To further explore the protective properties of XE991, retigabine was applied to another group injected with 6-OHDA and XE991. In this group, slower walking speeds, decreased balance beam scores, and increased grip time were noted compared to the group treated with XE991. The results of this analysis further support the neuroprotective capabilities of XE991, as activation of KCNQ channels via retigabine reverses the benefits of XE991 [80].

Although retigabine can reverse the XE991-related attenuation of motor symptoms in 6-OHDA treated rats, it has been hypothesized to diminish hyperkinetic motor symptoms caused by long-term treatment of PD. Considering that hypoactivity of dopaminergic neurons induced by PD-like states has been treated by the blockade of KCNQ channels via XE991, it would seem logical that activation of KCNQ channels via retigabine and other channel openers would attenuate symptoms due to dopaminergic hyperactivity seen in the treatment of PD. Currently, one of the primary treatment methods for PD includes the compound L-DOPA. Within 10 years of L-DOPA administration, many PD patients receiving this drug tend to become afflicted by hyperkinetic symptoms such as chorea, myoclonus, or dystonia [99,100,101]. In rat models of L-DOPA-induced dyskinesia (LID), it was found that retigabine significantly reduced the frequency of abnormal involuntary movements (AIM). However, it is important to note that retigabine was unable to completely prevent the occurrence of LID or delay the onset, even with chronic treatment. A limitation of retigabine as a pharmacotherapeutic option for the treatment of LID is its potential to induce sedation [102,103,104,105]. This effect, however, was reduced over the course of chronic treatment with retigabine. Similarly, the KCNQ2/3 opener, N-(6-chloro-pyridin-3-yl)-3,4-difluoro-benzamide (ICA 27243), was also found to significantly diminish the frequency of AIMs seen in rat models of PD [104,105]. It is hypothesized that both retigabine and ICA 27243 reduce AIM frequency by acting upon KCNQ2/3 in striatal projection neurons, whose increase in activity is associated with the symptoms of LTD [104,105].

As recent studies have shown, the modulation of dopaminergic activity via KCNQ channels has the potential to treat several motor disorders associated with both LID and Parkinsonian states. Due to the relative lack of side effects associated with drugs that modulate the activity of KCNQ channels, these compounds appear to be promising new avenues of pharmacotherapy in the management of PD. Further understanding of the therapeutic potential of these drugs is essential due to the side effects associated with current PD treatments, such as L-DOPA. A future direction should attempt to discern the existence of unknown adverse effects through continued trials of these drugs in PD murine models, as these drugs have only minimally been utilized in studies regarding the attenuation of motor symptoms. Additionally, future research into the ability of KCNQ modulators to treat other disorders related to dopaminergic pathways is another avenue that should be explored as well. Past studies have speculated that KCNQ openers, such as retigabine and ICA 27243, may be able to reduce the excitability of dopaminergic neurons in other tracts, such as the mesolimbic pathway, where overactivity results in schizophrenic symptoms [105,106,107,108,109,110,111,112,113]. This could be a potentially innovative tactic in clinical medicine as well, especially in the case of dementia-related psychosis, which is common onset among those with neurodegenerative disease. Moreover, recent clinical trials on the oral use of primavanserin, an atypical antipsychotic which has antagonist/inverse agonist effects on 5HT2A receptors, have been currently conducted. Within the most recent trial, a side effect of primavanserin use as a treatment for PD was QT prolongation [114]. This could be the particular area in which KCNQ channel modulation (e.g., retigabine), could aid in limiting these side effects. However, no current studies have yet to be carried out on this potential dual medication use. The KCNQ channel opening and closing mechanisms by selected agents are listed in Table 2. Overall, the ability of KCNQ channels to modulate a wide variety of motor-related dopaminergic pathologies is a promising field of study and may provide insight into the treatment of a wide variety of dopaminergic disorders in the future.

## 4. Discussion

The aim of this review was to encompass the updated literature and scope of KCNQ channel behavior in neural mechanosensation pathways. This aim was achieved by establishing the concept of VGICs, and the specific structure and physiology of the KCNQ family and related to both motor and sensory circuits. The novelty behind this updated review is that it encompasses relevant channelopathies as well as conceptual pharmacotherapeutic options via modulation of the KCNQ channel, as depicted in Figure 2. However, the future direction of KCNQ neural mechanosensation literature ought to focus more on the clinical and translational use of these pharmacotherapeutic options, as most of the literature used to support this review was based on animal models alone. Despite this, there is strong evidence that could support the initiation of future clinical trials, especially for auditory pathologies as well as in PD [14,80,94,95,96,97,98,99,100,101,102,103,104,105,115,116,117,118,119,120,121,122], and developments for understanding the effects that antimicrobial agents have on these potassium ion channels [123,124,125,126,127,128,129,130,131,132,133,134,135,136,137,138]. Additionally, the scope of this review was on neural modulation, yet there is a growing amount of literature on the behavior of KCNQ channels in cardiac modulations as well, especially in the cellular processes involved in arrhythmia [26,27,28,29,139,140,141,142,143,144,145,146,147,148,149].

## 5. Conclusions

Overall, the aim of this review was to encompass the current literature and scope of KCNQ channels. KCNQ channels remain an imperative area of pathophysiological study. Some of the most pertinent roles of these channels include their mechanosensation behavior in dopaminergic motor disorders, pain sensation, neural plasticity, and cranial sensory transduction (e.g., hearing, olfaction, vision). Future directions ought to include exploring these treatment modalities in clinical trials as well as understanding cellular processes with non-neural disorders just as well.

## Figures and Tables

**Figure 1 membranes-12-00499-f001:**
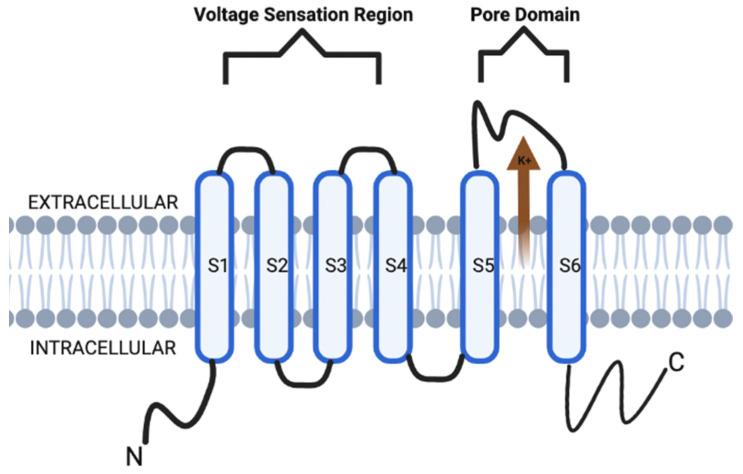
KCNQ channel structure is composed of six transmembrane segmental helices, classified as S1–S6. In addition, the S1–S4 transmembrane segmental helices compose the actual voltage sensation region, and the latter two (S5–S6) units are the actual gate of the channel.

**Figure 2 membranes-12-00499-f002:**
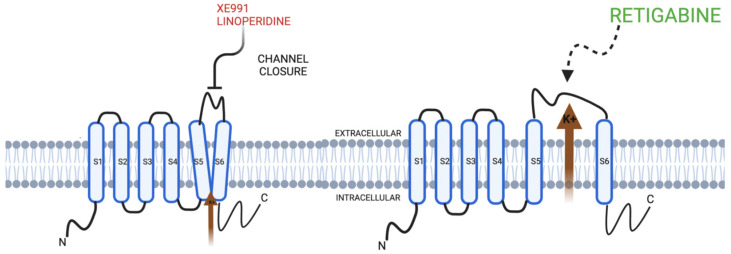
KCNQ channels and pharmacotherapeutic modulators in relation to the cellular membrane. Retigabine acts to open the KCNQ channel, whereas linopirdine and XE991 are channel inhibitors that act to inactivate the KCNQ channel function.

**Table 1 membranes-12-00499-t001:** Expression distribution and associated pathologies with channel genes.

Gene	Expression Distribution	Associated Pathologies
KCNQ1	Cochlea	Type 1 long QT syndrome
	Heart	
KCNQ2	Cerebellum	Benign familial neonatal seizures
	Hippocampus Medulla	Early onset epileptic encephalopathy
KCNQ3	Cerebellum	Benign familial neonatal seizures
	Hippocampus Medulla	Early onset epileptic encephalopathy Bipolar Disorder
KCNQ4	Cochlea Trigeminal ganglia	Deafness
KCNQ5	Retinal pigment epithelium	*

* No major associated pathologies. Of note, this table is not comprehensive to all expression and pathological distributions of these genes.

**Table 2 membranes-12-00499-t002:** Current Proposed Neuromodulation Tactics and Agents utilizing KCNQ Channels.

Action on Channel	Mechanisms	Agents
Opening	Phosphatidyl inositol 4, 5 biphosphate (PIP 2) Gamma-amino butyric acid (GABA) Inositol 1, 4, 5 triphosphate (IP3) a. PIP2 synthesis b. Suppression of muscarinic current (M) receptor at acetyl choline receptor	Retigabine Flupirtine ICA 27243
Closing	Muscarinic current (M) associated with acetyl choline receptor Pathological Mechanisms a. Movement Disorders	Linoperidine XE991

## Data Availability

Not applicable.

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
