# Peer review of "Behavior of KCNQ Channels in Neural Plasticity and Motor Disorders"

_membranes, 2022, doi:10.3390/membranes12050499_

Round 1

Reviewer 1 Report

In this manuscript, the authors summarized the current topics on KCNQ/Kv7 studies in neurons and also described the future perspectives on them. They are experts on acid-sensing ion channel, ASICs but not KCNQ channels. However, this manuscript includes recent insights on KCNQ channel in neurons.

Major concerns: Current figures are not impressive to increase the number of citations. Informative figures summarizing Section 2 and 3 (for each) should add. In addition, Figure or Table on clinical significance of KCNQ inhibitors/activators should add.

Author Response

In this manuscript, the authors summarized the current topics on KCNQ/Kv7 studies in neurons and also described the future perspectives on them. They are experts on acid-sensing ion channel, ASICs but not KCNQ channels. However, this manuscript includes recent insights on KCNQ channel in neurons.

Major concerns: Current figures are not impressive to increase the number of citations. Informative figures summarizing Section 2 and 3 (for each) should add. In addition, Figure or Table on clinical significance of KCNQ inhibitors/activators should add.

Response: Great point and thank you! Table 2 was created to outline KCNQ channel opening/closing mechanisms and agents which are discussed in this manuscript. In addition, both figures 1 and 2 have been modified extensively to show the subunits and channel regions (voltage sensing & pore domain). Figure 2 was further modified to include the subunit closing/opening actions when exposed to the agents discussed in Table 2. We believe this allows for the audience to have a visual supplement to understand the clinical significance these channels have in generate action potentials.

Reviewer 2 Report

In this review, the authors summarized the physiological and pathological function of KCNQ channels in neurons. Specifically, they talked about the role of KCNQ channels in the neuronal excitability and synaptic plasticity and how the channelopathies (motor disorders) are associated with dysfunctions of KCNQ channels. The manuscript seems interesting for some readers but not that exciting and comprehensive. Moreover, there are some major concerns that need to be addressed.

  1. English editing is highly recommended. Some sentences are extremely long and hard to read. For example, Line 32.
  2. Figures are not related to the texts and are not helping. Readers cannot see the actual gate, voltage-sensing domain and pore domain in Fig 1. Fig 1 is way too simple and needs to be remade.
  3. What are the roles of Na+-Ca2+ exchanger, TRP channels and Ca2+-ATPase in Figure 2? No need to show them if not talking about them in the manuscript.
  4. There is a huge redundancy between Discussion and Conclusion. Might only leave one in the manuscript.
  5. The manuscript is not well-organized. For example, the authors could work on a table listing all the openers and inhibitors of KCNQ channels and their effects on neural tissues by modulating KCNQ channels.

Author Response

In this review, the authors summarized the physiological and pathological function of KCNQ channels in neurons. Specifically, they talked about the role of KCNQ channels in the neuronal excitability and synaptic plasticity and how the channelopathies (motor disorders) are associated with dysfunctions of KCNQ channels. The manuscript seems interesting for some readers but not that exciting and comprehensive. Moreover, there are some major concerns that need to be addressed.

  1. English editing is highly recommended. Some sentences are extremely long and hard to read. For example, Line 32.

     Response: Thanks, and the grammar was extensively edited and long sentences were cut with help of native English speaker.

  1. Figures are not related to the texts and are not helping. Readers cannot see the actual gate, voltage-sensing domain and pore domain in Fig 1. Fig 1 is way too simple and needs to be remade.

     Response: Remade Figure 1 to show the subunits as described in “Introduction” section. In the previous iteration, there was a lack of vision in being able to see the subunits and regions of the channel. We believe our most modified figure allows for the audience to better understand the channel structure discussed in the introduction.

  1. What are the roles of Na+-Ca2+ exchanger, TRP channels and Ca2+-ATPase in Figure 2? No need to show them if not talking about them in the manuscript.

     Response: Great comments and we removed it from the Figure 2. This new figure is streamlined to focus on the use of KCNQ agents which open/close the channel. We hope this adds a degree of clarity to the manuscript.

  1. There is a huge redundancy between Discussion and Conclusion. Might only leave one in the manuscript.

     Response: Good point and we removed parts of conclusion to simplify overall synopsis of paper and future directions. In the conclusion, the overall aim of the review, salient applications of KCNQ channels (as discussed earlier in review), and a point on future directions for these channels were included to provide a “closure” for the manuscript.

  1. The manuscript is not well-organized. For example, the authors could work on a table listing all the openers and inhibitors of KCNQ channels and their effects on neural tissues by modulating KCNQ channels.

     Response: Great point. Table 2 was created to outline KCNQ channel opening/closing mechanisms and agents. Both figures 1 and 2 have been modified extensively to show the subunits and approach to closing/opening channels.

Reviewer 3 Report

The review by Singh, William, Malavia, and Chu has a large number of facts in 9 pages of text. The reader may benefit from additional summary tables. One example will be forwarded to the Editor as it would not fit into the constraints of this space. But that is just one such example. 

The authors might want to comment in lines 350-355, references 105-113 on Tariot et al Trial of Pimvanserin in Dementia-related psychosis. New England Journal of Medicine 2021; 385: 3099-319 in which prolonged QT interval on EKG was found as a side effect in treatment of a long-term complication of a movement disorder (Parkinson's Disease). 

Under abbreviations Line 424  6 ONDA Neurotoxin was not included. 

Author Response

The review by Singh, William, Malavia, and Chu has a large number of facts in 9 pages of text. The reader may benefit from additional summary tables. One example will be forwarded to the Editor as it would not fit into the constraints of this space. But that is just one such example. 

Response: Great point and it was shown in newly Table 2.

The authors might want to comment in lines 350-355, references 105-113 on Tariot et al Trial of Pimvanserin in Dementia-related psychosis. New England Journal of Medicine 2021; 385: 3099-319 in which prolonged QT interval on EKG was found as a side effect in treatment of a long-term complication of a movement disorder (Parkinson's Disease). 

Response: Comments were added in lines 354-362 with additional citation. This is a very interesting suggestion, and our team enjoyed integrating the potential connection and a future direction of KCNQ channel role in Pimvanserin-QT prolongation.

Under abbreviations Line 424,  6-OHDA Neurotoxin was not included. 

Response: Changed as suggested (see line 436).

Round 2

Reviewer 1 Report

I have no more concerns.

Reviewer 2 Report

The authors have addressed all my concerns.